# Selection signatures and population dynamics of transposable elements in lima bean

Daniela Lozano-Arce [1], Tatiana García[2], Laura Natalia Gonzalez-Garcia [1,3], Romain Guyot[3], Maria Isabel Chacón-Sánchez [2,4] & Jorge Duitama [1,4✉]

The domestication process in lima bean (*Phaseolus lunatus* L.) involves two independent events, within the Mesoamerican and Andean gene pools. This makes lima bean an excellent model to understand convergent evolution. The mechanisms of adaptation followed by Mesoamerican and Andean landraces are largely unknown. Genes related to these adaptations can be selected by identification of selective sweeps within gene pools. Previous genetic analyses in lima bean have relied on Single Nucleotide Polymorphism (SNP) loci, and have ignored transposable elements (TEs). Here we show the analysis of whole-genome sequencing data from 61 lima bean accessions to characterize a genomic variation database including TEs and SNPs, to associate selective sweeps with variable TEs and to predict candidate domestication genes. A small percentage of genes under selection are shared among gene pools, suggesting that domestication followed different genetic avenues in both gene pools. About 75% of TEs are located close to genes, which shows their potential to affect gene functions. The genetic structure inferred from variable TEs is consistent with that obtained from SNP markers, suggesting that TE dynamics can be related to the demographic history of wild and domesticated lima bean and its adaptive processes, in particular selection processes during domestication.

[1] Systems and Computing Engineering Department, Universidad de los Andes, Bogotá, Colombia. [2] Departamento de Agronomía, Facultad de Ciencias Agrarias, Universidad Nacional de Colombia, Bogotá, Colombia. [3] Institut de Recherche pour le Développement (IRD), UMR DIADE, Université de Montpellier, CIRAD, 34394 Montpellier, France. [4] These authors jointly supervised this work: Maria Isabel Chacón-Sánchez, Jorge Duitama. ✉email: ja.duitama@uniandes.edu.co

Lima bean (*Phaseolus lunatus* L.) is the second most important domesticated species of the genus *Phaseolus* after common bean (*Phaseolus vulgaris* L.). Wild populations of both species are distributed from Mexico to Argentina, presenting a wide range of ecological adaptations. For this reason, it is considered a promising crop to improve food security in predicted scenarios of climate change[1,2]. Four *P. lunatus* wild gene pools have been defined: two Mesoamerican (MI and MII) and two Andean (AI, AII)[3,4]. Different studies have shown that both, common bean and lima bean, have gone through at least two independent domestication processes[5]. Domesticated types of lima bean were mostly selected from Mesoamerican (MI) and Andean (AI) wild populations, and have been cultivated across the Americas since pre-Columbian times and in some African countries after Columbus. Although different research efforts have been performed to understand these domestication processes, the genetic drivers of adaptation during domestication remain largely unknown.

Recent progress in the development of high-throughput sequencing technologies has allowed the genome assembly of a large number of non-model species, increasing the genomic information for different crops[6]. Recently, Chacón-Sánchez et al. summarized the genomic resources generated in recent years within the *Phaseolus* genus, showing their importance to evaluate gene flow between gene pools and even between species[7]. Chromosome-level genome assemblies are available for common bean[8], tepary bean (*Phaseolus acutifolius* A. Gray)[9], and lima bean[4]. The lima bean genome was generated by sequencing of long reads from the MI accession (G27455) cultivated in northern Colombia. A second assembly, built from short reads, is composed of 19,316 scaffolds and belongs to the MI domesticated Bridgeton cultivar[10]. For the G27455 assembly, RNA-seq data from pod, leaf, and flower tissues were also generated, which complemented the transcriptome data generated as part of an assay evaluating resistance to the fungus *Trichoderma viride*[11]. Regarding intraspecies genetic diversity, Genotype-by-Sequencing (GBS) data is available for about 500 accessions of lima bean, covering the main pools of genetic diversity[3,4]. A recent study used 15,168 SNP markers from 183 lima bean accessions to evaluate the genetic consequences of introgressions and gene flow on the genetic structure and diversity of lima bean, focusing on the Yucatan Peninsula region[12]. Much knowledge can be gained from genomic data on poorly known aspects of the domestication process. For example, in lima bean we do not still know whether the genetic bases of the domestication syndrome, namely the morphological and physiological changes that differentiate wild and domesticated populations, are similar between the Mesoamerican and Andean domestication events.

A complete understanding of the evolution and diversity of crops requires the study of transposable elements (TEs). TEs are DNA sequences that have the ability to change their position within the genome in a replicative or non-replicative process[13]. TEs represent an important part of plant genomes, and in some cases comprise up to 80% of their total amount of DNA. Recent studies have shown that transposable elements are related to changes in the expression and function of genes in plants, thus playing an important role in their adaptive evolution[14–19]. Moreover, they are important drivers to the evolution of genomes, influencing processes such as speciation and selection during domestication[20–23]. One example in common bean is the report by Parker et al. of structural changes in the *INDEHISCENT* gene (*PvIND*) that control fiber loss or gain in pods[24]. These changes are due to a duplication of the locus and an insertion of a Long Terminal Repeat (LTR) retrotransposon (Ty1-copia), which are associated with overexpression of *PvIND* and loss of pod strings. Despite the importance of transposable elements, they have received little attention in the lima bean genome. Although an initial annotation of transposable elements was performed as part of the annotation of the lima bean genome, a detailed characterization and analysis of these elements has not been conducted in the same way it has been conducted for common bean[25]. In particular, the available information of genetic diversity is insufficient to identify and analyze intraspecies population dynamics of TEs. Given the high cost of performing sequencing and *de-novo* assembly of complete populations, whole-genome resequencing has been used to assess presence–absence variation of TEs in different crops[22,26].

In the present work we aim to identify and compare the genomic distributions of selective sweeps between the Mesoamerican and Andean gene pools and contribute to the study of the role of TEs in the evolution and adaptation mechanisms of lima bean during domestication. We present a curated annotation and a complete catalog of transposable elements in the genome of *P. lunatus*. Based on whole-genome resequencing of 61 accessions, we also built the most complete database of genomic variation for this species which was used to detect selective sweeps through multiple approaches. The analysis of this database also revealed genomic elements related to seed size and resistance to abiotic stresses. Moreover, we identified presence–absence variation related to population dynamics of TEs between and within the main gene pools of *P. lunatus*. Variable TEs in or close to genes are nominated as candidate drivers of traits related to domestication and breeding processes in lima bean.

## Results

**Improved catalog of transposable elements in lima bean and common bean.** We generated a new catalog of genome-wide transposable element (TE) annotations in the lima bean and the common bean genomes, using a combination of structure-based, homology-based, and *de-novo* methods. The TE annotation pipeline included the software tools Inpactor2[27], the Extensive *de-novo* TE Annotator (EDTA)[28], and RepeatMasker[29]. This pipeline produced a raw set of 621,418 TE annotations in the *P. lunatus* reference genome assembly, covering 308 Mbp (56.35%) of the lima bean genome assembly size. A large percentage (68%) of these TEs correspond to 99% of the TEs reported in the initial analysis presented in Garcia et al., for which only RepeatMasker was used (Supplementary Table 1)[4]. In both annotations, the raw dataset includes annotations classified as "Tandem" and "Unknown". Manual inspection of some of these events revealed that they did not correspond to TEs. Therefore, we removed 115,207 annotations classified as "Unknown" and 790 annotations classified as "Tandem" from the database. Conversely, the pipeline used in this work identified ten additional TE families from the DNA transposons group: DNA/DTA, DNA/DTC, DNA/DTH, DNA/DTM, DNA/DTT, MITE/DTA, MITE/DTC, MITE/DTH, MITE/DTM, and MITE/DTT. Although the pipeline identified a smaller number of TEs of the superfamilies LTR retrotransposons Copia and Gypsy, the total length of regions spanned by the new LTRs is larger than that obtained in the previous report. The reason for this outcome is that the new annotations correspond to complete LTRs, whereas many previous annotations were fragmented. Furthermore, the new pipeline provided subclassification into lineages for these LTRs.

Although for common bean Gao et al. reported a 2.12Mbp database containing 791 representative TE sequences distributed in 14 families[25], a genome-wide annotation of TEs was not available. Therefore, to compare the lima bean results against common bean, we also executed the same pipeline on the common bean genome. The pipeline identified a total of 580,817 TEs covering 48.50% of the genome assembly size. In this case we

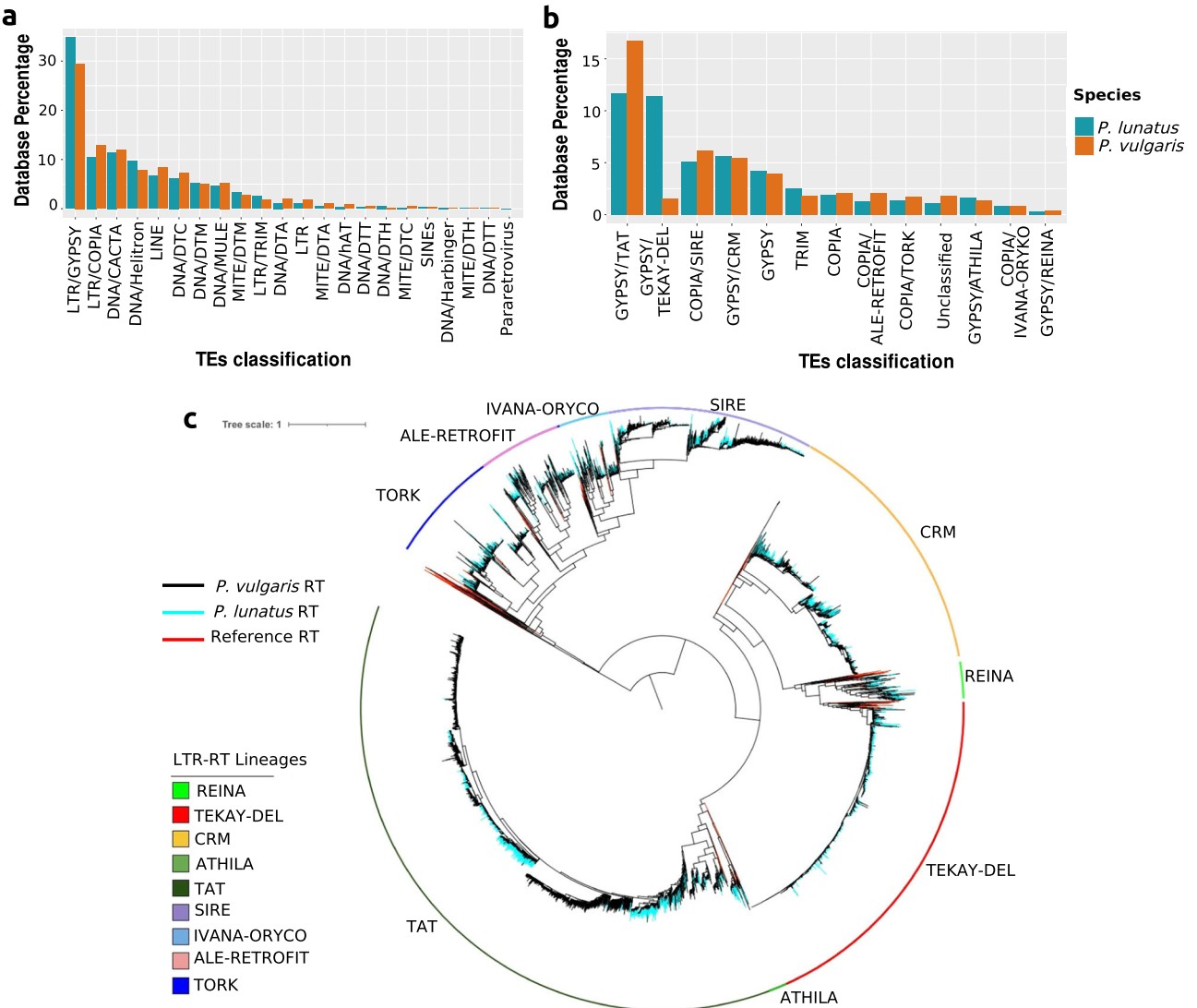

**Fig. 1 Transposable elements in lima bean and common bean. a** Superfamilies of TEs and **b** Gypsy and Copia LTR lineages present in the built databases for each species, *P. lunatus* (lima bean) and *P. vulgaris* (common bean). **c** Phylogenetic analysis and comparison of the *P. vulgaris* and *P. lunatus* LTR retrotransposon sequences encoding the reverse-transcriptase (RT) domains. The unrooted phylogenetic tree of Gypsy (REINA, CRM, TAT, ATHILA, and DEL-TEKAY) and Copia (TORK, ALE-RETROFIT, IVANA-ORYCO, and SIRE) elements includes 5312 *P. lunatus* (blue) and 4264 *P. vulgaris* (black) aligned sequences (longer than 200 amino acids). The red lines indicate reference RT domains used to determine the clades.

also removed 113,076 TEs classified as "Unknown" and 224 classified as "Tandem". Similar to the lima bean annotation, the pipeline used in this work annotated DNA and MITE DNA transposon families, as well as LTR retrotransposon lineages, which were not identified in previous analysis.

The initial TE sequences identified were filtered according to quality criteria based on the length distribution for each family (see "Methods" for details). This allowed us to identify and classify a total of 223,780 TEs in the lima bean, which cover 254 Mbp (46.5% of the assembly, Supplementary Data 1). The most representative superfamilies are LTR/Gypsy (34.81%), followed by DNA/CACTA (11.47%) and LTR/Copia (10.55%) (Fig. 1a, Supplementary Table 1). Likewise, a total of 230,300 TEs were annotated in the common bean reference genome, spanning 218 Mbp (41.8% of the assembly, Supplementary Data 2). The order of the three most representative families was also LTR/Gypsy (29.44%), LTR/Copia (12.90%), and DNA/CACTA (11.87%). LTR transposons in both species are mainly composed of the Gypsy and Copia autonomous families, and the TRIM (Terminal Repeat in Miniature) non-autonomous families. Figure 1b shows

the distribution of families and lineages within the Gypsy and Copia superfamilies. The main difference between the two bean genomes is the abundance of GYPSY/TAT and GYPSY/TEKAY-DEL lineages showing an increment of the TAT subclade in the *P. vulgaris* genome, and an increased number of the TEKAY-DEL subclade in the *P. lunatus* genome.

To further understand the diversity of LTR retrotransposons, a phylogenetic reconstruction was performed using the reverse-transcriptase (RT) domains. Figure 1c shows the distribution of diversity of LTR subclades: Gypsy (REINA, CRM, TAT, ATHILA, and TEKAY-DEL) and Copia (TORK, ALE-RETROFIT, IVANA-ORYCO, and SIRE), combining LTRs of *P. lunatus* and *P. vulgaris* (See independent trees in the Supplementary Figs. 1 and 2). As observed in the distribution of percentages (Fig. 1b), there is a recent expansion of LTR/Gypsy RT domains of the lineage TEKAY-DEL in *P. lunatus* after the divergence from the common ancestor with *P. vulgaris*. The combined tree also shows a group of LTRs of the TAT lineage that are not present in *P. lunatus*, which also suggests a recent expansion of these domains within the *P. vulgaris* genome. The diversity of LTRs of the remaining

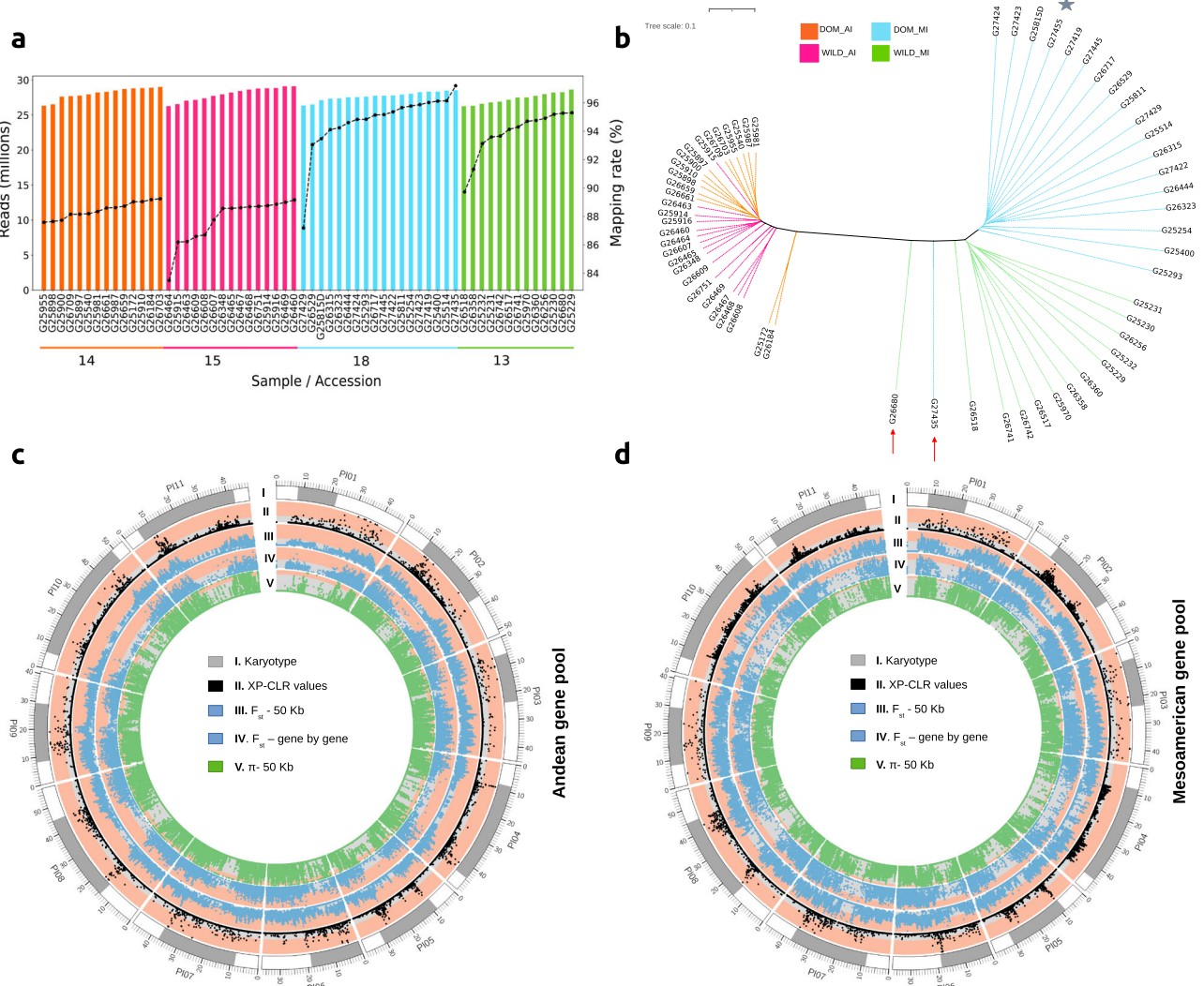

**Fig. 2 Genomic variation and selection signatures. a** Number of WGS Ilumina reads obtained for 61 sequenced accessions. The black dotted line indicates the percentage of reads aligned against the lima bean reference genome. Colors differentiate the population of origin for each accession (DOM_AI=domesticated Andean, WILD_AI= wild Andean, DOM_MI= domesticated Mesoamerican, WILD_MI= wild Mesoamerican). **b** Neighbor joining clustering of samples based on SNP genotype calls. Colors differentiate the population of origin for each accession. The accessions highlighted with a red arrow (G27435, G26680) are admixed between Mesoamerican gene pools (MI and MII) and the accession marked with a star corresponds to the reference genome. **c**, **d** Comparison of selected windows identified by the XP-CLR approach, by $F_{ST}$ indexes and reduction in nucleotide diversity ($\pi$) in 50 Kb/5 Kb sliding windows, and by the gene-by-gene approach in **c** the Andean gene pool and **d** the Mesoamerican gene pool. Salmon colors indicate the region of significance for each statistic.

subclades is evenly distributed between species, which suggests that these elements were inserted before the speciation process.

**Genomic variability and signatures of domestication in lima bean gene pools**. To explore the genetic diversity within lima bean, we performed Illumina whole-genome resequencing on 60 *P. lunatus* accessions, including wild and cultivated accessions of the MI and AI gene pools (Supplementary Data 3). Over 25 million paired-end reads were sequenced for each accession, targeting an average read depth over 10x (Fig. 2a). The mapping rate for all accessions against the *P. lunatus* reference genome was greater than 83% and the lowest percentages were observed in wild AI accessions.

We assembled a raw variation database including 7,316,508 SNPs. The number of genotype calls different from the reference allele is consistent with the population of origin of each sample (Supplementary Fig. 3). The AI accessions have between two and four times the number of variants compared to wild MI and

domesticated MI accessions. The domesticated MI (G27435) and wild MI (G26680) accessions, showing the largest number of variants within their population, were previously classified as admixed between the gene pools MI and MII[4]. The minor allele frequency (MAF) distribution of the overall population, derived from the raw genotype calls, shows an excess of SNPs with high frequency of the minor allele (Supplementary Fig. 4). This can be explained by the population structure of the sequenced samples. Filtering by MAF, observed heterozygosity and minimum number of individuals genotyped, we obtained a curated database of 1,724,831 SNPs, which we used for downstream analysis. A neighbor joining tree, obtained from genetic distances between the sequenced samples, shows a clear differentiation of the AI, wild MI, and domesticated MI populations (Fig. 2b). This tree is consistent with the study by Garcia et al., in which Genotype-by-sequencing (GBS) data was generated for 482 accessions[4].

Two sliding-window-based approaches and a gene-based approach were applied to the curated genomic variation database

to identify and compare the genomic distribution of selective sweeps in wild and domesticated lima bean accessions within each gene pool (AI and MI). Results are summarized in Fig. 2c, d.

In the first sliding-window approach, we used the cross-population composite likelihood ratio test implemented in XP-CLR[30] on windows of 50 Kb/5 Kb to identify selective sweeps as those regions with extreme allele frequency differentiation among wild and domesticated populations within each gene pool. As a result, we predicted selective sweeps for 1182 genes in the Andean gene pool and 1278 genes in the Mesoamerican gene pool (Supplementary Data 4). Chromosomes Pl01, Pl03, Pl07, Pl09 and Pl11 included over 100 genes with selective sweeps in the Andean gene pool (Supplementary Table 2). In the Mesoamerican gene pool more than 100 genes with selective sweeps were also found in chromosomes Pl02 and Pl08. A total of 236 genes were shared between gene pools.

In the second sliding-window approach, genomic data were evaluated across 50-Kb/5-Kb sliding windows with the PopGenome program[31]. Within each gene pool, reduction in nucleotide diversity in the domesticated accessions (measured as ($\pi_{wild}$ - $\pi_{domesticated}$)/$\pi_{wild}$ ratios) and $F_{ST}$ indexes among wild and domesticated accessions were calculated for each window. Selective sweeps were identified as those windows in the top 10 percent of the distribution of both low diversity and high differentiation values. For the Andean gene pool, we predicted selective sweeps for a total of 2263 genes, while for the Mesoamerican gene pool we identified 2007 sweeps (Supplementary Data 4). Although these numbers were larger than those obtained using XP-CLR, only 202 genes were shared between gene pools.

For the gene-based approach, we calculated the nucleotide diversity and $F_{ST}$ on each gene in the gene catalog of the lima bean genome to detect candidate genes under selection. We selected the genes in the top 10% of the distribution of low genetic diversity and high $F_{ST}$. With this approach, we predicted selective sweeps for 694 genes in the Andean gene pool and 981 genes in the Mesoamerican gene pool (Supplementary Data 4). Only 29 genes were shared among gene pools. For the Andean population, chromosomes Pl2, Pl03, Pl07, and Pl09 had the highest gene counting, while for the Mesoamerican gene pool, chromosomes Pl02, Pl03, Pl04, and P07 showed the highest number of genes with signatures of selection (Supplementary table 2).

The previous approaches generated important information about likely selective sweeps in lima bean. Therefore, we evaluated different combinations of the individual results to select a gene subset with a high probability of being in regions under selection. The intersection of the three approaches resulted in only 58 and 93 genes in the Andean and Mesoamerican gene pools, respectively (Supplementary Figs. 5 and 6). Within the Andean gene pool the response to photo-oxidative stress (GO:0080183) biological process was enriched in genes selected by the second approach. The oxidoreductase activity (GO:0016899) molecular function was enriched in genes selected by the first approach. Finally, the protein kinase complex (GO:1902911) cellular component was enriched in genes selected by the three approaches (Supplementary Fig. 7). These GO categories suggest the likely relation of the gene set involved in adaptation to changing light environments through the control of photo-oxidative stress. Several oxidoreductase enzymes participate in photosynthetic electron transport to chloroplast redox metabolism[32]. In the Mesoamerican gene pool, ontologies related to metabolism of $1,3-\beta-D-glucan$ across the main categories (molecular function: GO:0003843; Biological process: GO:0006075, GO:0006074, GO:0051274; cellular component: GO:0005774) were enriched in the subset of 264 genes selected by the two window-based approaches (Supplementary Fig. 8). This metabolite is a polysaccharide found in a wide variety of

plants, fungi, and bacteria as the main component of primary cell walls. In plants, it is synthesized in different development stages and tissues, especially in pollen mother cell walls and pollen tubes. Also, $1,3-\beta-D-glucan$ plays a role in a range of abiotic and biotic stresses due to their accumulation between the plasma membrane and the cell wall after exposure of plants to stress conditions[33]. For instance, in common basil (*Ocimum basilicum L.*) Alhasnawi evaluated the reduced negative effects of salt stress in plants under β-glucans treatments[34]. Likewise, Liang et al. reported the direct relation of overexpression of NAC transcription factor in oat with the content and biosynthetic of (1,3;1,4)-β-D-glucan, which improves salt and drought tolerance[35].

According to the significant GO enrichment results, we used the genes selected by both sliding-window-based approaches (XP-CLR and popgenome) within each gene pool to carry out the Kyoto Encyclopedia of Genes and Genomes (KEGG) pathway analysis (Supplementary Figs. 9 and 10). Within the Andean gene pool the analysis selected pathways regarding carbohydrate metabolism, which are important photosynthesis products and source for several plant processes as growing cells. Within the Mesoamerican gene pool the analysis selected the mitogen-activated protein kinase (MAPK) signaling pathway, which plays a role in activating and signal transduction in several abiotic stress conditions (salt, cold, and drought)[36]. One interesting gene belonging to this pathway is *Pl04G0000254700*, which is a VP1-transcription factor involved in plant stress responses in *A. thaliana*[37] and *B. napus*, where also it is considered a hub gene together with MYB44 in responses to drought and salt stresses[38].

Interestingly, we found two genes related to traits in the domestication syndrome in selective sweeps in both gene pools. The first is the gene *Pod Dehiscence 1* (*PDH1*) (*Pl03G0000340600*) which is responsible for the lignin deposition in the wall fiber layer of the pod, and contributes to splitting the pod valves[39,40]. In wild common bean, fibrous and strongly lignified cell layers differ in pod anatomy compared to landraces where fiber layers are reduced[41]. The second gene is *Pl07G0000312000* (*P Locus*) which has been associated with seed color[42]. In the Mesoamerican gene pool, we further identified a gene set involved in response to stress conditions. The first is gene *Pl11G0000087300* (PIP2), which belongs to the aquaporins protein family, it is involved in water permeability in vacuolar and plasma membranes and plays an essential role in drought resistance[43]. Besides, the homologous gene *Phvul.011G079300* was previously reported by Schmutz et al. as a target of selection in Mesoamerican common bean[8]. The second gene is *Pl06G0000180000* (ASN1), whose overexpression leads to higher nitrogen and seed soluble protein content and increased seed weight[42]. *Pl04G0000029100* (ABCB19) is an ABC transporter protein involved in several processes related to plant architecture in common bean, such as apical dominance and hypocotyl gravitropism[42].

We also compared the genes detected under selection in lima bean with those reported by Schmutz et al. in common bean that were associated with domestication[8]. In the common bean dataset, 1835 genes were identified in the Mesoamerican gene pool and 748 genes were identified in the Andean gene pool, for a total of 2524 genes (only 59 genes were observed in common). We identified 2361 orthologs of these genes in lima bean, 1726 in the Mesoamerican gene pool, 686 of the Andean gene pool, and 51 shared between gene pools. From these, 428 (24.8%) and 168 (24.4%) were also included in lima bean selective sweeps by at least one approach and in at least one gene pool (Supplementary Data 4). These numbers reduce to about half if the intersection is performed separately for the two regions of domestication.

From the genes selected in the Mesoamerican gene pools of both species (common and lima bean), we highlight five genes supported by all approaches (*Pl01G0000337700.v1*,

*Pl06G0000127200.v1*, *Pl11G0000105700.v1*, *Pl11G0000106200.v1* and *Pl11G0000120200.v1*). Functional characterization of those genes shows us relation with compounds that play a role in several plant defense pathways such as lectins coded by the *Pl01G0000337700.v1* gene, which together with receptor-like kinases (RLKs) and receptor-like proteins (RLPs) generates a plant response to different biotic and abiotic stimuli[44]. Also, the gene *Pl11G0000105700.v1* is involved in the production of ferredoxin proteins in the chloroplast, and it participates in the redox regulation process and antioxidant defense in plants[45]. The gene *Pl11G0000106200.v1* codes for a protein transport *SEC24-1* and *Pl11G0000120200* codes for a beta-1,3-galactosyltransferase, which has been reported in *Arabidopsis* to have an important role in seedling development especially in the micropylar endosperm[46].

**Presence–absence variation (PAV) related to population dynamics of transposable elements**. Understanding the potential importance of TEs as genetic drivers of the phenotypic variation that was selected during the domestication processes, we also analyzed the WGS data to provide information on the composition of the lima bean mobilome, i.e., the dynamics of TEs within the species. To identify potential deletion events spanning annotated transposable elements, we ran the functionality to identify large deletions available in NGSEP from paired-end reads with abnormally large predicted fragment lengths[47]. A presence–absence variation matrix was derived from deletions called in individual accessions, checking for each accession and for each annotated TE whether at least 85% of the TE overlapped with a deletion event. After filtering low quality calls (see "Methods" for details), 39,459 TEs were identified as having evidence of deletion in at least one accession (Supplementary Data 5). These deletion events involved the full range of Class I LTR and non-LTR retroelements (i.e., GYPSY, COPIA, and LINE superfamilies) and Class II DNA transposons (i.e., hAT and CACTA superfamilies).

Figure 3a shows the counts of TE deletion events (relative to the reference) for each sample, adding up to a total of 332,758 individual deletion events (Supplementary Table 3). The counts of these deletions allowed us to differentiate the two Andean (AI) and Mesoamerican (MI) populations (Supplementary Fig. 11). These counts were compared with a Wilcoxon rank test between each pair of populations (DOM_AI, WILD_AI, DOM_MI, WILD_MI). Significant differences were observed for all combinations, with the exception of the comparison between WILD_AI - DOM_AI and WILD_MI - DOM_MI. (Supplementary Table 4). Similar to the SNVs database, the MI/MII admixed accessions G27435 and G26680 had the highest count of deletion events among the Mesoamerican accessions. A neighbor joining clustering of the PAV matrix differentiates the AI and MI populations and most of the accessions between the domesticated and wild Mesoamerican (MI) populations (Supplementary Fig. 12).

Although the minor allele frequency (MAF) distribution of variable TEs does not have the peak close to 0.5 observed in the distribution derived from SNP variability (Supplementary Fig. 13), 1653 TEs with PAV differentiate the Andean and Mesoamerican gene pools (Fig. 3b, Supplementary Data 6, Fisher test *p*-value < $10^{-10}$). These TEs might be insertions that occurred in the Mesoamerican population at least $0.5010 \pm 0.02611$ million years before present (mybp) when the divergence between lima bean gene pools likely occurred[48]. The LTR/Gypsy superfamily is overrepresented in this group with 1439 TEs (87%; *p*-value = 2.2 e-16 of a chi-square test). Comparing wild and domesticated Mesoamerican accessions, there were no TEs differentiating these

gene pools at p-value < $10^{-10}$. However, 61 TEs have a significant difference in allele frequencies at a *p*-value < $10^{-5}$ (Supplementary table 5). We hypothesize that these TEs have been selected by the domestication process within the MI population. Finally, 9326 TEs correspond to singleton deletions, which suggests recent deletion of these elements[26].

**Genomic variation related to domestication genes**. As reported by previous studies, TE dynamics can impact gene function and expression if TE insertions occur in the proximity or within coding regions[18]. To assess the potential impact of TEs on lima bean genes, we selected TEs located in the vicinity of protein coding genes, and having PAV within the sequenced accessions. From the 39,459 PAV TEs, 29,824 were observed in the flanking regions of about 38% of the genes in the transcriptome, at a 10 Kbp window. Comparing gene pools, a larger number of variable TEs close to genes were observed in the Andean gene pool compared to the Mesoamerican gene pool, probably because the reference genome was sequenced from a domesticated Mesoamerican accession. A larger number of variable TEs close to genes were observed in wild populations, compared to domesticated populations in both gene pools (Supplementary Fig. 14).

Focusing on genes previously reported as important in the domestication process in lima bean[4,41,49,50], we found variable TEs associated with 13 genes related to traits of pod shattering, cyanogenesis, photoperiodic flowering, flowering time, growth habit, drought tolerance, 100 seed weight, and plant architecture (Supplementary Data 7). Figure 4a, b highlights the case of the *GIGANTEA* (*GI*) gene, which codes for the unique plant-specific nuclear protein. Many pleiotropic functions in various physiological processes have been reported for this gene, such as regulation of flowering time, light signaling, starch accumulation, chlorophyll accumulation, transpiration, herbicide tolerance, cold tolerance, and drought tolerance[50]. In soybean, a mutant haplotype of this gene has been associated with early flowering time in cultivated genotypes[51]. Consistent with the inversion previously reported between lima bean and common bean in chromosome Pl04[4], the coding strand of the orthologous gene in common bean (*Phvul.004G088300*) is the negative strand of the reference genome, whereas the coding strand for the lima bean gene (*Pl04G0000200500*) is the positive strand of the reference genome (Fig. 4a). We identified 29 TEs associated with the common bean gene, in a 10 Kbp window upstream and downstream of the gene. For lima bean we recorded 28 associated TEs, five of them having evidence of variability. The first variable TE is an LTR/Copia/ALE-RETROFIT (4063 bp) upstream of the gene. In the second intron we identified two variable TEs of LTR/Gypsy/TAT and in the third intron we identified another two variable TEs of the same classification.

Figure 4b shows the reference allele and seven alternative alleles observed in the population, taking into account the PAV identified for the five TEs. The reference allele is carried by 41 accessions (allele I). Six Mesoamerican accessions (four wild and two domesticated) carry an alternative allele, which misses the TE within the first intron (allele VIII). This is the only non-reference allele present within the Mesoamerican gene pool. Within the Andean gene pool, the LTR Copia located before the transcription start site is missing in two domesticated accessions (allele III). All intron TEs are present in these accessions. One wild accession misses the four intron TEs (allele II), whereas three domesticated accessions and one wild accession only retain the smallest intron TE (allele VII). Alleles IV, V and VI, represented in four domesticated and two wild accessions show different configurations missing one or two intron TEs.

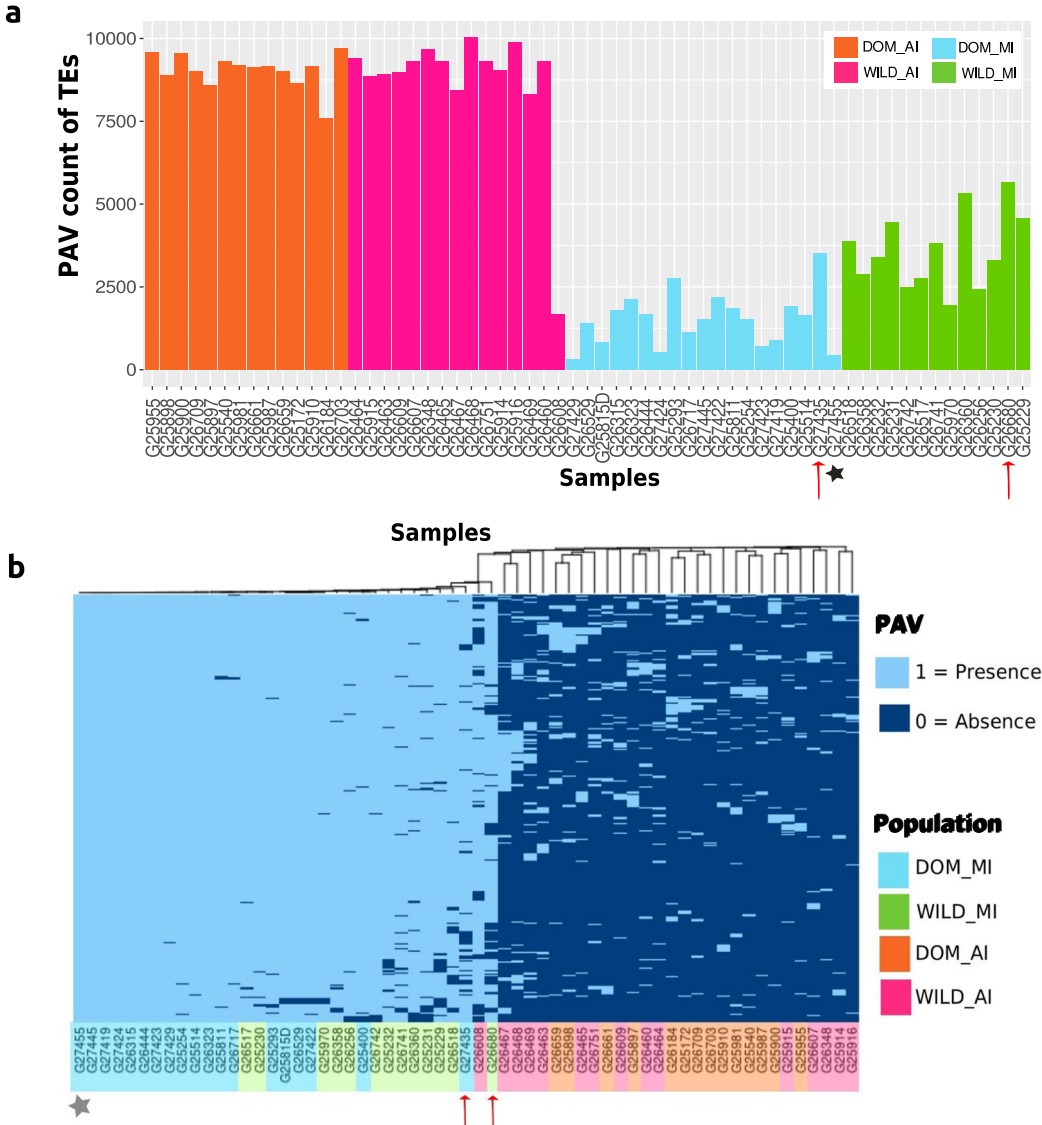

**Fig. 3 Differentiation of gene pools through variable transposable elements. a** Count of Presence–absence variation (PAV) of annotated TEs for each lima bean accession. **b** PAV alleles of variable TEs mostly present in Mesoamerican accessions and absent in most Andean accessions. Colors differentiate the population of origin for each accession (DOM_AI=domesticated Andean, WILD_AI= wild Andean, DOM_MI= domesticated Mesoamerican, WILD_MI= wild Mesoamerican). The accessions highlighted with a red arrow (G27435, G26680) are admixed between Mesoamerican gene pools (MI and MII) and the accession marked with a star corresponds to the reference genome.

TEs also have been targets of evolutionary processes such as domestication, allowing their rapid fixation in the selective sweeps. To further investigate adaptive evolution in lima bean domestication, mediated by TE insertion events, we detected those PAV TEs that occurred within selective sweeps identified with SNP markers. We found 22,639 TEs located in selective sweep regions identified in the present study. Of those, 11,331 TEs were present exclusively in sweep regions within the Mesoamerican gene pool, 11,308 TEs were localized in selective sweeps within the Andean gene pool, and 6471 TEs were identified in both gene pools. An interesting case of a variable TE affecting a selected gene was identified in the Mesoamerican gene pool in the *Pl04G0000100000* gene in lima bean (Fig. 4c). This gene belongs to the subfamily of aquaporins known as intrinsic plasma membrane proteins (PIPs). Aquaporins are involved in many plant physiological processes, such as cell differentiation and elongation, plant transpiration, and regulation of plant hydraulics[52]. Several studies have shown that aquaporins are related to the response to drought stress in common bean varieties, and marked differences in gene expression were observed in drought-resistant versus susceptible genotypes[53–55]. We identified five TEs with variability in the lima bean gene (*Pl04G0000100000*) in the fourth and sixth introns (Fig. 4c). Besides, 22 additional TEs are located upstream and downstream of this gene. *Phvul.004G082700* is the aquaporin homologous gene in common bean. In this gene, 13 associated TEs were found, only one of them located within the first intron. TE insertion explains the longer genomic span of this gene in the lima bean genome, compared to the common bean genome. Figure 4d shows the allelic variation within the population, based on the variable TEs. In this case, only one non-reference allele was identified in which the five variable TEs are not present. This allele appears in four (about 25%) wild Mesoamerican accessions and in one wild Andean accession. Conversely, the reference

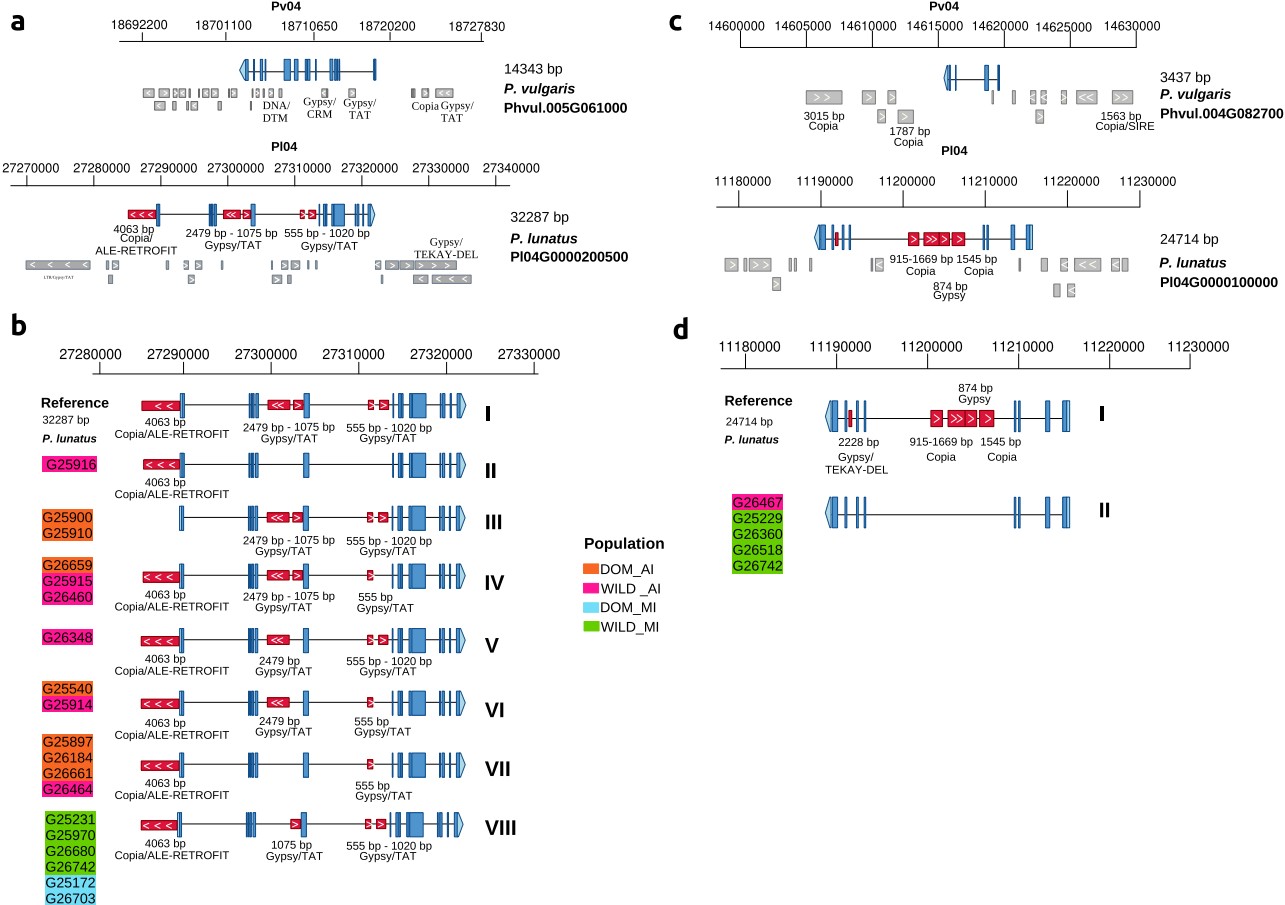

**Fig. 4 Diversity of presence/absence variation of TEs within genes. a** Gene model of the *GIGANTEA* gene in common bean and lima bean, including annotated TEs. TEs with (PAV) are colored red. TEs without variability are colored gray. **b** Representation of the alleles of the *GIGANTEA* gene of lima bean, according to PAV of TEs. The accessions highlighted with colors have the corresponding non-reference allele. **c** Gene model of the *AQUAPORIN* gene in common bean and lima bean, including annotated TEs. TEs with (PAV) are colored red. TEs without variability are colored gray. **d** Representation of the alleles of the *AQUAPORIN* gene of lima bean, according to PAV of TEs. The accessions highlighted with colors have the corresponding non-reference allele.

allele including the five TEs is fixated in the domesticated populations, supporting the reduction in diversity expected for a selective sweep.

## Discussion

The *Phaseolus* genus represents a unique example of multiple and parallel domestication. Lima bean and common bean developed similar genetic structures through their independent domestication events, making interspecific comparison between them possible[56,57]. In this study, we performed WGS of wild and domesticated populations to identify selective sweeps through the lima bean genome that could be used to annotate genomic elements related to the domestication processes occurring in the evolutive history of lima bean. Given the growing evidence indicating that dynamics of transposable elements is a main driver of phenotypic variation in plants[15–23], we decided to take into account both protein coding genes and TEs in our investigation of genomic elements related to selective sweeps. The analysis of high-density SNP markers and TE presence–absence polymorphisms proved useful not only to identify genomic regions affected by selective sweeps, but also to identify regions in which TE polymorphisms could have been the target of selection, an information that would be missed with the use of SNP markers alone. A similar approach has been applied recently in genetic diversity of tomato to improve the significance of genotype-phenotype associations[26].

As a baseline for this work, we identified, classified, and annotated TEs in the lima bean and the common bean genomes using a combination of homology, structure, and *de-novo* approaches, including deep learning approaches. As a result, we generated a curated database of transposable elements for *P. lunatus*, including 223,780 TEs and covering 254 Mbp of the genome assembly (about 46.5%). Although a database of 791 non-redundant transposons was previously generated for common bean[25], a complete annotation of TEs throughout the genome was not available. This becomes a practical limitation to analyze the dynamics of TEs between species. Hence, we also developed a TE database for the *P. vulgaris* genome that includes 230,300 TEs and covers 218 Mb of the genome assembly size (about 41.8%). The annotation and re-annotation of the common bean and the lima bean genomes allowed the identification of new families, notably in the DNA and MITE groups and an identification at the lineage level for the LTR retrotransposon group. The improved detection and annotation of TEs is mainly due to the continuous improvement of bioinformatics pipelines, combining robust methodologies. The identification and classification of novel MITES (or Miniature Inverted-repeat Transposable Elements, non-autonomous class II elements) appears particularly interesting for future studies. Indeed, MITEs are often inserted in the vicinity of genes where they can play a role in the regulation of gene expression and promoting mutations[58–60].

The improvement in this database was crucial to understand the TE dynamics between lima bean and common bean. A phylogenetic analysis of the RT domains of LTR retrotransposons allowed us to identify important differences in TE dynamics between lima bean and common bean. These differences were likely to occur after the split of the two species from their most recent common ancestor. The Gypsy/TEKAY-DEL lineage shows very recent proliferation in lima bean as evidenced by the large number of short branches in phylogeny. On the other hand, the Gypsy/TAT lineage shows diversification and differential proliferation of subgroups between lima bean and common bean. This independent proliferation of LTR retrotransposons can be induced by biotic and abiotic stresses and as a consequence, can cause a sudden increase in genome size as observed for *Oryza australiensis*[61]. The Gypsy/TEKAY-DEL lineage is particularly active in several plant species and represents a significant fraction of their genome[62]. Recently, a comparative analysis in three *Capsicum* species showed a significant variation in the proportion of the Gypsy/TEKAY-DEL lineage[63] demonstrating their propensity to accumulate rapidly in genomes. Their recent amplification in lima bean will provide interesting markers to understand their dynamism and impact at the intraspecies level. Indeed, it has been reported that the insertion of transposable elements could be a factor of innovation and adaptation to a changing environment[64]. Especially in species that have widely expanded their geographical range, as in wild and domesticated lima beans, populations had to adapt to a variety of ecological and agro-ecological conditions. Considering that domestication in lima bean was a very recent event, early domesticates had to adapt rapidly to new selection pressures driven by humans and TEs may have contributed to this adaptation, as it is shown below.

The analysis of WGS data from 61 wild and domesticated *P. lunatus* accessions, allowed us to investigate at the same time selective sweeps and intraspecies TE dynamics. As a first milestone towards this goal, we generated the first dense genetic variability database, including genotypic information for 7,316,508 SNPs. As expected, the global analysis of this variability database was consistent with that obtained from genotyping by sequencing data in previous studies[3,4]. Moreover, the assembled database allowed a nearly complete reconstruction and analysis of genetic variation for individual genes. This database is a resource of genetic markers for future breeding activities. Moreover, this database is a main resource to identify selective sweeps related to domestication processes in lima bean. Combining different approaches, we identified selective sweeps in up to 10% of the gene models. Regardless of the identification method, less than 12% of the genes with selective sweeps were shared among gene pools, suggesting that domestication may have been achieved in both gene pools by different genetic avenues. A similar result was observed in common bean where only 2.3% of genes identified under selection were shared among gene pools[8]. Interestingly, we found that more than 500 genes observed under selection in lima bean were also detected in common bean, suggesting that a group of genes might have been consistently selected in the domestication of both species. Comparing gene pools, we obtained more genes related to selective sweeps in the Mesoamerican pool than in the Andean pool with the gene-based approach. This could suggest a faster protein evolution within the Mesoamerican gene pool. However, this result could be produced by bias generated by the fact that the reference genome was assembled from an accession with Mesoamerican origin. Moreover, further experiments are needed to evaluate the contribution of standing genetic variation and new beneficial mutations in the response to selection during domestication. Also, this distinction is critical if one aims to understand how similar phenotypes arise from independent domestication events, as it has occurred in lima bean.

Based on several recent studies on different species, it could be argued that dynamics of transposable elements can play a more important role in the genomic structure and the phenotypic variation of species, compared to SNV mutations[15–23]. Two of the main effects caused by TE insertions are the regulation of gene expression through cis or trans elements in TE sequences, and the generation of epigenetic modifications caused by TE insertions or deletions[65,66]. The identification of large deletions from paired-end WGS data allowed us to characterize some of the TE dynamics occurring within lima bean variability. We built a catalog of presence–absence variation (PAV) of TE to generate the first TE mobilome in lima bean. We acknowledge that the use of short reads limits the number of TE variation events that could be correctly identified and genotyped at the population level. Nevertheless, following a conservative approach, we could assess presence–absence allelic variation for 39,459 TEs. Some of these variable TEs differentiate the Andean and Mesoamerican gene pools, as well as wild from domesticated populations within the MI gene pool. Because our analysis is guided by a Mesoamerican reference genome, we identified fewer PAVs within Mesoamerican accessions compared to more distant Andean accessions. The MAF distribution of PAVs did not show the high frequency peak observed in the MAF distribution derived from SNPs. A possible explanation for this behavior is that structural variants tend to be deleterious and hence they might be subject to negative selection, which produces an excess of low frequency alleles. Nevertheless, the genetic structure inferred from PAV TEs agrees with that obtained from whole-genome SNPs. Therefore, these TE events can be related to the demographic history of wild and domesticated lima bean and its adaptive processes, in particular to the selection processes during domestication. A large number of genes related to different processes contain variable TEs within intronic regions. Although it could be argued that these insertions should not have an important effect in gene function because they are "synonymous" regarding gene products, evidence from other plants suggests that some of these insertions could alter gene expression through different mechanisms. As an example, about 10% of the maize genes have at least one intronic TE insertion, and some of these insertions have been associated with high levels of CHG methylation and dimethylation of lysine 9 of histone H3 (H3K9me2), playing a role in chromatin modifications[67]. These results suggest the importance of characterizing methylation patterns in lima bean in future research.

The differences observed in gain/loss of TEs among wild and domesticated accessions in the Mesoamerican gene pool of lima bean may be due to the fact that domestication may have initially involved few genotypes, thus contributing to increased divergence among wild and landrace populations due to the effects of genetic drift. Also, the presence/absence of some of these TEs, especially those close or within genes, may have provided some advantage to landraces and therefore may have been unconsciously selected in favor by early farmers. For example, it has been reported that in maize a transposon located between 58.7 Kbp and 69.5 Kbp upstream of the gene (*tb1*) was an enhancer of gene expression, which explains the differences in plant architecture between maize and its wild relative[68]. That study showed how TEs can be a means of rapid adaptation, since they can quickly create genetic diversity in addition to being enhancers of gene expression[69]. It is interesting to note that of the 39,459 PAV TEs observed in lima bean, 22% were located in intergenic regions and 75% were located close to or within genes, thus providing a great potential of TEs to affect gene functions in lima bean. To further explore the role of TEs in domestication, we show intraspecies variability of TEs in proximity to genes previously related to domestication and agronomic traits such as sheath dehiscence, cyanogenesis, and flowering time. While this study marks a good starting point,

future studies should increase the availability of whole-genome assemblies and WGS data on a larger set of wild and domesticated accessions.

Given the importance of lima bean as a current food security crop, we believe that both the databases and the compiled information reported here will provide a basis for future studies on the evolution and function of TEs in different plant species, as well as applications to genetic improvement of lima bean. In particular, in the short term, we expect to build genome assemblies of different accessions including the Andean gene pool (AI).

## Methods

**Reference genomes**. The *P. lunatus* V.1 reference genome and the *P. vulgaris* V.1.0 were retrieved from Phytozome v.13. These genomes were used as a baseline for TEs identification, classification and annotation. The lima bean genome has a total length of 546.42 Mbp[4]. The common bean genome has a total length of 520.99 Mbp[8].

**Identification, classification, annotation and filtering of transposable elements**. The assembled genomes (*P. lunatus* V.1 and *P. vulgaris* V.1.0) were used to identify transposons as follows. Inpactor2[27] was used to identify complete LTR transposons using a machine learning approach. Then, to identify TEs by similarity, the previously identified TE sequences in *P. vulgaris*[25] were clustered with the Inpactor2 detected sequences using CD-HIT[70]. One TE sequence per cluster was retained based on length and the expected set of domains in the family. This filtered database was used to generate the specific classification by superfamilies using the Extensive *de-novo* TE Annotator (EDTA)[28]. Due to the strict filters used by the RepeatMasker[29] step in the EDTA pipeline, the RepeatMasker analysis was repeated using the EDTA library as input. This procedure allowed us to integrate homology and structure signals in the classification process to complete the annotation and characterization of the catalog of transposable elements of the *P. lunatus* genome (Supplementary Fig. 15). The regions that were annotated with both software tools (RepeatMasker and EDTA) were separated into superfamilies. Unknown annotations, simple repeats, low complexity regions, tandem repeats, and annotations of pseudogenes were discarded. For each superfamily, TEs were filtered out to remove small size annotations (See Supplementary Table 6 for more detail). Finally, redundant annotations were merged, thus reducing the number of elements initially annotated.

**Phylogenetic reconstruction of LTR transposons lineages**. Phylogenetic trees were reconstructed using the retrotranscriptase domain of LTR transposons as was previously described[71]. First, each genome was compared against a RT database using CENSOR[72] retaining RT domains with a minimum length of 150 amino acids. This reference database was composed of the GypsyDB[73] and the REXdb[74] databases. Mapping results were filtered by 50% identity and 50% alignment length. Then, the identified RT domains and the reference domains were concatenated into a final RT database. All RT domains were aligned using MAFFT (v. 7.475)[75] and an approximated maximum likelihood phylogenetic tree was reconstructed using FastTree (v. 2.1.11)[76] and edited with Itol[77].

**Illumina whole-genome sequencing of 60 accessions**. We performed whole-genome sequencing of 60 accessions obtained from the International Center for Tropical Agriculture (CIAT) (See Supplementary Data 3 for details). This included 32 domesticated accessions (14 from the Andean AI gene pool and 18 from the Mesoamerican MI gene pool) and 28 wild accessions (15 from the AI gene pool and 13 from the MI gene pool). Young trifoliate leaves from two-week-old seedlings were collected and frozen with liquid nitrogen. Based on the DNA integrity and concentration requirements of Illumina sequencing technology, DNA extraction was performed using the extraction method developed by Vega-Vela & Sánchez[78]. The Illumina library used 1.0 μg of DNA according to a NEBNext DNA Library Preparation Kit following the manufacturer's recommendations (New England BioLabs, Ipswich, MA, USA). Genomic DNA was fragmented to 350 bp in size, fragments ligated to NEBNext adapters and enriched by PCR. Library size distribution was analyzed with an Agilent 2100 Bioanalyzer (Agilent Technologies, Santa Clara, CA, USA) and quantified by real-time PCR. Libraries were sequenced on an Illumina HiSeq platform (Illumina, San Diego, CA, USA) using a 150 paired-end run (2 × 150 bases) and an insert size of 450 bp. Raw WGS data is available at the NCBI sequence read archive database with bioproject accession number PRJNA596114.

**Detection and genotype of single nucleotide polymorphisms (SNP)**. Illumina reads from all 60 accessions sequenced (WGS) were quality validated and mapped to the *P. lunatus* reference genome using Next Generation Sequencing Experience Platform (NGSEP) V.4.2[79]. Illumina reads from the reference genome accession (G27455), were retrieved from NCBI (SRR10726092) and were also mapped to the lima bean reference genome (Supplementary Fig. 16). Variants were identified and individuals were genotyped using the MultiSampleVariantDetector command from the NGSEP V.4.2[79] with the following parameters: -maxAlnsPerStartPos 2 as maximum number of alignments allowed to start on the same reference site, -maxBaseQS 30 as the maximum value allowed for a base quality score, -h 0.0001 as the heterozygosity rate (prior probability of finding a heterozygous SNP at each position) and -knownSTRs with the File with known lima bean short tandem repeats (STRs). A raw set of reliable variants was obtained by filtering with the NGSEP VCFFilter command with the following criteria: -q 40 minimum genotype quality score (coded in Phred, where 40 means 0.9999 posterior probability of that each genotype call is correct), and -frs to remove repetitive regions. This procedure generated a set of 7,316,508 biallelic SNVs with approximately 33% missing data.

This initial variation database was further filtered to exclude variants with minor allele frequency (MAF) < 0.05, variants with maximum observed heterozygosity >0.1, and to retain only biallelic SNPs. Variants with less than 40 genotyped samples were also discarded. A total of 1,724,831 SNPs was obtained after this step. This database was used to reconstruct a tree topology for genetic diversity analysis based on the Neighbor-Joining (NJ) approach. The VCFDistanceMatrixCalculator and NeighborJoining commands from the NGSEP were used. The tree was visualized and edited with iTOL v.4.4.2103[77].

**Identification of selective sweeps**. To identify selective sweeps, we applied an integrative approach focusing upon three approaches that compared wild and domesticated accessions within each gene pool: (1) a likelihood method based on the calculation of a multilocus allele frequency differentiation statistics between populations applied to genomic sliding windows. (2) evaluation of diversity and genetic differentiation indexes ($\pi$ and $F_{ST}$) by a genomic sliding-window approach. (3) evaluation of diversity indexes by a gene-by-gene approach. The identification of selective sweeps was done on the filtered SNP database consisting of 1,724,831 SNP loci.

In the first approach, we evaluated allele frequency differentiation at linked loci among wild and domesticated accessions within each gene pool with the statistics called XP-CLR (cross-population composite likelihood ratio test)[30] on windows of 50 Kbp / 5 Kbp. Selective sweeps were identified as those windows with XP-CLR normalized values ≥ 5. In the second approach, genomic data were evaluated across 50 Kbp / 5 Kbp sliding windows with the PopGenome program[31]. Within each gene pool, reduction in nucleotide diversity in the domesticated accessions (an effect known as founder effect) (measured as ($\pi_{wild}$ - $\pi_{domesticated}$)/ $\pi_{wild}$ ratios) and $F_{ST}$ indexes among wild and domesticated accessions were calculated for each window. Selective sweeps were identified as those windows in the top 10 percent of the distribution of both low diversity and $F_{ST}$ values. In the third approach, we applied the same criteria as in the second approach to detect domestication candidate genes in Mesoamerica and the Andes. For this, we calculated allele sharing diversity statistics (the average number of pairwise differences per Kbp and $F_{ST}$) through all the genes in the catalog of the lima bean genome using the VCFAlleleSharingStats module of NGSEP and selected the genes within the top 10 percent of the distribution of low diversity and $F_{ST}$ values. The distribution of selective sweeps was compared among the Mesoamerican and Andean gene pools of lima bean and also to the genomic regions potentially affected by selective sweeps that have been previously identified in common bean[8]. Finally, we generated a consensus on the results obtained by all the approaches to obtain a list of candidate genes that can be validated in future studies.

**Identification of presence–absence variability (PAV) of TEs**. The Single Sample Variants Detector from NGSEP software V.4.2[79] was run independently for each sample, activating the read pair analysis to identify large deletions from paired-end reads. Then, presence/absence variation of TEs was inferred from the overlap between the TE location and the deletions identified by NGSEP (Supplementary Fig. 17). For each accession and each TE, the TE was considered as deleted (allele zero) within the accession if at least a fixed percentage of the base pairs of the TE overlap with a deletion event. Otherwise, the reference allele was coded with the number one. Four different matrices of genotyped TEs were obtained according to the percentage of the transposon that was deleted (100%, 95%, 90%, and 85%). We selected the 85% matrix for the following analysis according to experimental data. Briefly, we calculated the number of variable TEs using the four minimum percentages (Supplementary Table 7). According to this metric, the 85% matrix presented the largest number of TEs with PAV (52,276). Filtering out TEs with length <500 bp, we obtained a total of 39,459 PAVs (Supplementary Data 5).

The PAVs were clustered using Hierarchical Clustering. The hclust package was used to create the clusters and plot.hclust to visualize the results in R v. 4.1.3. The matrix was transformed to a VCF format, and the VCFDiversityStats command of NGSEP was used to calculate minor allele frequencies (MAF) of the variable TEs.

In order to detect TEs that differ in frequency between the Mesoamerican and Andean gene pools, and also between the wild and domesticated accessions within the Mesoamerican gene pool, we applied fisher exact tests to each TE (Supplementary Data 5). Subsequently, with this matrix, the ComplexHeatmap package[80] of R was used to visualize associations between different TEs between the collections.

**Identification of variable TEs in genes related to domestication**. The annotated genes of *P. lunatus* were contrasted with the TE database to identify associated transposons. A 10 Kbp window was used to identify TEs upstream and downstream of the genes. From the comparison file, the search for genes related to domestication was performed. Subsequently, with the JBrowse web, the genome was navigated and the insertions of TEs near or within the genes were visualized. To validate the presence/absence variation (PAV) of a TE (size 1423 bp) annotated in the *GIGANTEA* gene, seven accessions from different gene pools (Mesoamerican, Andean, wild and domesticated) were randomly selected. The Samtools V.1.10 software[81] was used to extract the fraction of the mapping file (BAM file) corresponding to the TE and 100 kb flanking regions: "Pl04:27199747-27401170". Afterwards, each region was indexed, and visualized with the Integrated Genome Browser (IGV) tool[82]. Finally, the orthologous genes in *P. vulgaris* V.1.0 and V.2.0 were identified using the GenomesAligner command from the NGSEP V.4.2[83]. The position of the genes was extracted and visualized in JBrowse.

**Statistics and reproducibility**. This study analyzes Illumina whole-genome sequence data from 61 lima bean accessions. For comparisons between populations, the number of individuals per population was 13 for the wild MI population, 19 for the domesticated MI population, 15 for the wild AI population, and 14 for the domesticated AI population. Biological seed for all accessions used in this study can be requested to the International Center for Tropical Agriculture (CIAT). The methods to map reads to the reference genome, to build the genomic variation databases of SNPs and variable TEs, and to filter these databases are fully described in the corresponding methods sections. Full details on the statistical methods to identify selective sweeps are described in the section "Identification of selective sweeps". Significance of gene ontology enrichment within selective sweeps was assessed running the Fisher exact test available in the software TopGO v4.3[84].

To determine significance of differences from the PAV counts of TEs between gene pools (Andean and Mesoamerican) and biological status (wild and domesticated) a Wilcoxon test was carried out using the R function Wilcox.test with the alternative "two.sided". An exact Fisher test was used to assess significance in PAV allele frequencies between populations (MI vs AI and wild MI vs domesticated MI). Finally, a chi-square test was performed running the R chisq.test function on the counts of annotated TE superfamilies to assess overrepresentation of each particular family in the genomes of *P. lunatus* and *P. vulgaris*. The number of data points per category for this test was always larger than 137 (Supplementary Table 1).

**Reporting summary**. Further information on research design is available in the Nature Portfolio Reporting Summary linked to this article.

## Data availability

The data used in this study is available at the NCBI sequence read archive (SRA) database (https://www.ncbi.nlm.nih.gov/sra) with bioproject accession number PRJNA596114. The genomic variation database is available at the European Variation Archive (EVA) database under the bioproject accession number PRJEB62157. The reference genome assembly is available at the Assembly database of NCBI (https://www.ncbi.nlm.nih.gov/assembly/) with the accession number GCA_013389735.1. The genome is also available on Phytozome (https://phytozome-next.jgi.doe.gov/). Annotated transposable elements are included as supplementary files of this publication (Supplementary Data 1 and 2). Numerical source data for graphs and charts is available (Supplementary Data 8). All other data are available from the corresponding author (or other sources, as applicable) on reasonable request.

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

## Acknowledgements

The work presented in this manuscript was supported by internal funding of Universidad de los Andes through the FAPA research fund and a project to tackle the goals of sustainable development, awarded to JD. We also acknowledge the DSIT high performance computing unit at Universidad de los Andes for their support to conduct the analyses presented in this manuscript. The authors acknowledge the IFB Core Cluster that is part of the National Network of Compute Resources (NNCR) of the Institut Français de Bioinformatique (https://www.france-bioinformatique.fr). RG thanks the BIO_ANDES LMI for support.

## Author contributions

J.D. and M.I.C.S. conceived the study. M.I.C.S. performed field and lab work to sequence the samples. D.L.A., L.N.G.G. and R.G. performed bioinformatic analysis of transposable elements. D.L.A., T.G., and J.D. performed analysis of WGS data. All authors contributed to write the manuscript and approved the final version.

## Competing interests

The authors declare no competing interests.
