## [Peer Review File · Communications Biology]

Reviewers' comments:

Reviewer #1 (Remarks to the Author):

In the study by Lozano-Arce et al, the *Phaseolus lunatus* and *vulgaris* genomes were analyzed for transposable elements, with the goal of better understanding TEs and their structure and dynamics in adaptation and convergent evolution in the *Phaseolus* species. The authors also deliver an extensive genomic dataset of both TEs in these two species, and whole genome sequencing in 61 *P. lunatus* accessions. This study represents a significant and high quality amount of work, although all of the bioinformatic techniques used are outside of my expertise. The comparisons of transposable elements in these two species to my knowledge, has not been attempted before and given they seemed to evolve in similar areas of the world, I agree with the authors that these are good samples to use for convergent evolution. My comments, both minor and more major (marked by *), are listed below. Please note that some of the major comments are more subjective and my assessment after reading this manuscript several times.

Abstract:

Line 24: spell out WGS

Results:

Lines 119-122: In these two sentences, I am not entirely sure I understand the process. Do you mean that the 115,207 contained the low complexity and simple repeat regions found by Garcia et al (2021), and then you went on to identify an additional 10 families?

*Lines 123-125: here, I feel that a little more information could be helpful; is it possible to include a spreadsheet of the families there were identified in this study from the Garcia et al data, as well as the 10 additional families, with a brief description of what these families are? I understand that transposable elements are a very specific topic for a specific audience, but to make this of wider interest, it would be helpful to briefly explain these families, and to list somewhere (supplementary file?) what they are. Right now, the significance/meaning of these findings is not really coming through.

Line 131: change 115.207 to 115,207

Line 132: it might be my own lack of knowledge, but indicating that DNA and MITE were included in this list makes me wonder if they usually are not? Could you please provide some explanation of why you made this statement?

Lines 228 – 230: Does this statement about selective sweeps indicate that the MesoAmerican gene pool is gaining more beneficial mutations quicker than the Andean gene pool? If this is the case, then what might be some reasons/significance of this? I know that the authors go into more detail on a subset in the next several pages (see my comment below). Out of curiosity, are any of the other genes (outside of the smaller subset) involved in resistance and/or clustered on the chromosomes?

Pages 7 – 12 and figure 2: After a lot of reads and re-reads, this section, to me, is almost a different paper. The way it reads to me, is that we depart from TEs and move into alleles that are under pressure in the two species. I like the follow-up discussion in lines 479-490, and collectively, these two sections to me are interesting, but seem like an outlier in terms of the overall goal the authors present in the title and in the abstract. This whole idea of selective sweeps is cool, and there is perhaps enough data to move it into another paper. But I leave it to the editors and authors to figure this out. For me, it sort of detracts from the TE story, or is not tied in well enough to the TE story. Perhaps it's a matter of changing the title, making a better segue between lines 24 and 25 in the Abstract, and reorganizing this section/moving some to the Discussion section.

Discussion:

Line 472: here, adaptation to changing environment is discussed... do the authors have any thoughts on how this applies to lima bean in changing conditions? Does this make sense in terms of the lima bean environments during their radiation?

Missing from this section is discussion about a statement they made based on Figure 1B, which to me, read as important in the Results section. This statement is on 145-147, and discusses the main differences between the two genomes being the abundance of Gypsy/TAT and Gypsy/TEKAY Del. But these are not really discussed further. It seems like an important distinction between the two genomes.

Reviewer #2 (Remarks to the Author):

In the study, the authors identified regions in the genome of *P. lunatus* under artificial selection that were affected during the two domestication events that occurred in this species. Furthermore, they investigate the role that the TEs might have had in the domestication and adaptive processes in the two gene pools (from Mesoamerica and the Andean region).

The methods and approaches are adequate for the goals of this work. The results and data generated are an important contribution to our understanding of mostly unexplored sources of genetic variation, such as TEs and how they might contribute to adaptation and domestication.

My comments about the manuscript:

-Lines 119-121: The sentence regarding the TEs previously reported is not clear to me.

-Figure 1: Please be consistent with the names of the TEs. There are some differences in the names between 1B, the tree and the legend of the tree.

-Figure 1C: The colours that indicate the LTR-RT lineages in the tree do not entirely match the colour of the legend. Improving the colour would make the figure easier to interpret.

-Lines 182-184 and Supp Fig. 4: The pattern observed in the SFS can be affected by the MAF filter, the population structure and the different number of accessions per gene pool. This is because the MAF filter was applied to the complete dataset and not according to the gene pools. In order to see the pattern of the SFS, the MAF filtering should be removed or performed for each gene pool. Having the expected SFS would help to improve the interpretation of the SFS.

-Lines 186-188: I suggest applying another phylogenetic approach to construct the tree, such as the maximum-likelihood method used for the LTR transposons (FastTree).

-Fig 2C-D): What does the salmon colour indicate? Are the values in the salmon outliers?

-Line 228: low genetics diversity and high F_{st} ?

-Lines 350-352: Why a peak close to 0.5 is expected? Adding the expected SFS would allow comparison if there is a deviation between the observed and expected pattern of the SFS. There are neutral and non-neutral evolutionary processes that can be affecting the SFS.

-Lines 367-368: What percentage of the *P. lunatus* genes presented TEs in their flanking regions?

-Lines 579-581: What do the numbers after "domesticated-" and "wild-" mean? This is not clear to me.

Dear reviewers

Many thanks for your evaluation of our manuscript 'Selection signatures and population dynamics of transposable elements in Lima bean'. We performed additional data analysis and made the changes in the manuscript needed to address each comment. Please find our answers below for each specific comment. To facilitate the revision process, we marked in red the changes performed from the previous version of the manuscript.

Reviewers' comments:

Reviewer #1 (Remarks to the Author):

*In the study by Lozano-Arce et al, the Phaseolus lunatus and vulgaris genomes were analyzed for transposable elements, with the goal of better understanding TEs and their structure and dynamics in adaptation and convergent evolution in the Phaseolus species. The authors also deliver an extensive genomic dataset of both TEs in these two species, and whole genome sequencing in 61 P. lunatus accessions. This study represents a significant and high quality amount of work, although all of the bioinformatic techniques used are outside of my expertise. The comparisons of transposable elements in these two species to my knowledge, has not been attempted before and given they seemed to evolve in similar areas of the world, I agree with the authors that these are good samples to use for convergent evolution. My comments, both minor and more major (marked by *), are listed below. Please note that some of the major comments are more subjective and my assessment after reading this manuscript several times.*

R. We thank the reviewer for the careful assessment of our manuscript. We are glad to know that our work was perceived as a significant contribution to the field. We made changes to the results and discussion to address each comment of the reviewer. In particular, we improved the results and discussion to provide a more comprehensive view of the different results obtained in this work. Please find below our specific answer to each comment

Abstract:

Line 24: spell out WGS

R. We spelled the acronym as whole genome sequencing.

Results:

Lines 119-122: In these two sentences, I am not entirely sure I understand the process. Do you mean that the 115,207 contained the low complexity and simple repeat regions found by Garcia et al (2021), and then you went on to identify an additional 10 families?

R. We improved the writing of these sentences to clarify the new process to filter the raw set of TEs generated by Inpactor2, EDTA and Repeat Masker. From the raw dataset we removed 155,207 predicted TEs that were classified as "Unknown" and 790 that were classified as "Tandem". We performed manual inspection of some of the regions spanned by these

annotations to verify that their structure does not correspond to a transposable element. These annotations are usually included in raw TE catalogs because Repeatmasker is a generic tool to identify repeats. Conversely, the use of EDTA allowed us to annotate and classify 10 additional families (DNA/DTA, DNA/DTC, DNA/DTH, DNA/DTM, DNA/DTT, MITE/DTA, MITE/DTC, MITE/DTH, MITE/DTM, and MITE/DTT).

**Lines 123-125: here, I feel that a little more information could be helpful; is it possible to include a spreadsheet of the families there were identified in this study from the Garcia et al data, as well as the 10 additional families, with a brief description of what these families are? I understand that transposable elements are a very specific topic for a specific audience, but to make this of wider interest, it would be helpful to briefly explain these families, and to list somewhere (supplementary file?) what they are. Right now, the significance/meaning of these findings is not really coming through.*

R. We followed this suggestion and improved the supplementary table 1 with a summary of the comparison between the TE annotation performed by Garcia et al. 2021 and the annotation described here. We improved the text of the results and discussion to explain the importance of the new families identified in this analysis.

Line 131: change 115.207 to 115,207

R. We updated the number according to the latest results and fixed the number format

Line 132: it might be my own lack of knowledge, but indicating that DNA and MITE were included in this list makes me wonder if they usually are not? Could you please provide some explanation of why you made this statement?

R. The main reason to add this statement was that the current catalog of TEs for common bean (Gao et al. 2014) did not include elements of some DNA families and MITEs. It could be discussed if MITEs should be annotated separate from DNA transposons because formally, MITEs are derivatives of DNA transposons (or TIRs, Wicker et al., 2007). However, they can be considered as different genomic objects, because of their specific structure, length, distribution and the tools used to identify them. As mentioned in the previous comment, we improved the results and discussion to highlight the importance of the new families identified in this study. Studies in different plant species show that MITEs have a close physical association with genes in plants and are important in gene regulation and genome evolution (See for example Lu et al., 2012).

Lines 228 – 230: Does this statement about selective sweeps indicate that the MesoAmerican gene pool is gaining more beneficial mutations quicker than the Andean gene pool? If this is the case, then what might be some reasons/significance of this? I know that the authors go into more detail on a subset in the next several pages (see my comment below).

R. Unfortunately we can not draw this conclusion directly from the number of genes related to selective sweeps. We included a few sentences in the discussion with a brief summary of the following reasons. First, we have as a technical caveat that the reference genome is sequenced from a Mesoamerican accession, which could produce some level of bias in the number of discovered genes. Our research groups are currently conducting a new study which includes sequencing and assembly of an Andean accession, which we can use to validate this hypothesis.

More importantly, because domestication likely occurred in Lima bean within the last 6,000 years, populations in Mesoamerica and the Andes had to adapt quickly to new selection pressures carried by the domestication process. One could therefore conceive that the number of selective sweeps we observe in domesticated populations may not only be related to the sudden occurrence of new beneficial mutations, but also to neutral or nearly neutral alleles (from standing genetic variation) that became the targets of selection after environmental change. Because domestication entails big and complex changes in the environment, it is reasonable to think that these changes could have differed in the two domestication processes, not only in their nature but also in their frequency. On the other hand, the rate at which populations gain new beneficial mutations depend on multiple factors such as effective population size (determined by demographic history), mutation rate, selection coefficients, among others. In summary, the higher number of genes with selection signatures that we observed in Mesoamerica compared to the Andes might be the result not only of a higher rate of new beneficial mutations but also of a higher rate of selection pressure on alleles from standing genetic variation.

If the interest is to establish whether the Mesoamerican gene pool is gaining more beneficial mutations quicker than the Andean gene pool, we should be able to distinguish between selective sweeps on new beneficial mutations from those on standing genetic variation. This distinction is much easier to do when selection is strong on new beneficial mutations and when the frequency of “pre-domestication” alleles is high in wild populations. Because domestication traits (for example, loss of seed shattering) are usually disadvantageous for wild populations, the frequency of “pre-domestication” alleles is thought to be low in wild populations, thus making this distinction difficult to do. To establish whether domestication relied more on new beneficial mutations than on standing genetic variation is a fascinating subject that deserves further investigation in Lima bean in future studies.

Out of curiosity, are any of the other genes (outside of the smaller subset) involved in resistance and/or clustered on the chromosomes?

R. In the previous work on the Lima bean genome, Garcia et al. 2021 reported 1,917 genes related to resistance focusing on LRR (leucine-rich repeat-containing), toll/interleukin-1 receptor (TIR), leucine zipper (LZ), coiled-coil (CC), nucleotide-binding site (NBS/NB) shared by ARC (Apaf-1, R proteins, and CED-4) (NBS/NB-ARC) domain, serine–threonine kinase, and WRKY. Of those, we identified 391 genes related to resistance in selective sweeps. For instance, in

chromosome PI06, three genes present a NB-ARC domain (PI06G0000157000.v1, PI06G0000157400.v1, and PI06G0000157800.v1).

Pages 7 – 12 and figure 2: After a lot of reads and re-reads, this section, to me, is almost a different paper. The way it reads to me, is that we depart from TEs and move into alleles that are under pressure in the two species. I like the follow-up discussion in lines 479-490, and collectively, these two sections to me are interesting, but seem like an outlier in terms of the overall goal the authors present in the title and in the abstract. This whole idea of selective sweeps is cool, and there is perhaps enough data to move it into another paper. But I leave it to the editors and authors to figure this out. For me, it sort of detracts from the TE story, or is not tied in well enough to the TE story. Perhaps it's a matter of changing the title, making a better segue between lines 24 and 25 in the Abstract, and reorganizing this section/moving some to the Discussion section.

R. We improved different parts of the manuscript (abstract, results and discussion) to improve the cohesiveness of the results presented in this study. Here is a summary of our rationale to include the results in a single manuscript. On one side, we have been working for several years on the identification of selective sweeps in common bean and Lima bean following different approaches. Here, we report for the first time in Lima bean the construction of a dense genomic variation database and its use to identify and analyze selective sweeps. On the other side, there is a growing amount of evidence on the role of TEs and their mobility as drivers of rapid variation for several traits in plants. It could even be argued that TE dynamics can be more relevant than point mutations in protein coding genes. Hence, we believe that the investigation of genomic elements within the selective sweeps should include not only the protein coding genes, but also the TEs annotated close to these genes. Combining these ideas, we believe that TE dynamics and selective sweeps should be presented together to generate comprehensive hypotheses on the genomic elements that were selected during domestication.

Discussion:

Line 472: here, adaptation to changing environment is discussed... do the authors have any thoughts on how this applies to lima bean in changing conditions? Does this make sense in terms of the lima bean environments during their radiation?

R. Previous work in our research group has shown that wild Lima bean originated in the northern Andes (Ecuador-northern Peru) about 1 million years ago (Serrano-Serrano et al., 2010). After its origin, wild Lima bean expanded its range to Mesoamerica (up to northern Mexico) and to the southern Andes (up to northern Argentina). During this expansion, the major mesoamerican and andean gene pools formed and wild Lima bean had to adapt to a variety of ecological conditions, some of them driven by geological events such as the major orogeny of the Andes, which implied adaptation to higher elevations. Besides, Lima bean was domesticated about 6,000 years ago as mentioned earlier. In the domestication areas in Mesoamerican and the Andes, ancestral populations had to adapt rapidly to new environments under selection pressures mostly driven by humans. Also, humans expanded the early domesticates out of its area of origin to other regions in Mesoamerica and South America where

they had to adapt to new agroecological conditions. Due to their wide ecological range, wild Lima beans and their domesticated counterparts may be potential reservoirs of alleles for several important agricultural traits such as tolerance to high temperatures or other kinds of abiotic and biotic stresses. Some of these adaptations might have been reached by transposable elements generating or truncating regulatory elements, inducing gene mutations or chromosome rearrangements. We added to the discussion a couple of sentences summarizing our thoughts about this subject.

Missing from this section is discussion about a statement they made based on Figure 1B, which to me, read as important in the Results section. This statement is on 145-147, and discusses the main differences between the two genomes being the abundance of Gypsy/TAT and Gypsy/TEKAY Del. But these are not really discussed further. It seems like an important distinction between the two genomes.

R. Although this result was already discussed (lines 455-472 of the original text or third paragraph in the revised version), we believe that the connection to the result was missing because we were referring to the Gypsy/TEKAY-DEL family just as Del. We updated the references to this family in the text to improve the clarity and the connection with the result in figure 1B.

Reviewer #2 (Remarks to the Author):

*In the study, the authors identified regions in the genome of *P. lunatus* under artificial selection that were affected during the two domestication events that occurred in this species. Furthermore, they investigate the role that the TEs might have had in the domestication and adaptive processes in the two gene pools (from Mesoamerica and the Andean region). The methods and approaches are adequate for the goals of this work. The results and data generated are an important contribution to our understanding of mostly unexplored sources of genetic variation, such as TEs and how they might contribute to adaptation and domestication.*

R. We thank the reviewer for the assessment of our work. We are glad to know that the results are considered as an important contribution. We followed the suggestions of the reviewer performing the requested analysis and changes in the manuscript. Please find below our specific answers to each comment.

My comments about the manuscript:

-Lines 119-121: The sentence regarding the TEs previously reported is not clear to me.

R. We looked back at the sentences in this paragraph, which were pointed out by both reviewers, and noticed that there were many ideas mixed in this and other sentences. We improved the writing of the entire paragraph.

-Figure 1: Please be consistent with the names of the TEs. There are some differences in the names between 1B, the tree and the legend of the tree.

R. We apologize for the confusion with names. We changed the names of the TEs in figure 1 and in the section to make everything consistent.

-Figure 1C: The colours that indicate the LTR-RT lineages in the tree do not entirely match the colour of the legend. Improving the colour would make the figure easier to interpret.

R. We updated the figure 1C to improve the consistency of the colors.

-Lines 182-184 and Supp Fig. 4: The pattern observed in the SFS can be affected by the MAF filter, the population structure and the different number of accessions per gene pool. This is because the MAF filter was applied to the complete dataset and not according to the gene pools. In order to see the pattern of the SFS, the MAF filtering should be removed or performed for each gene pool. Having the expected SFS would help to improve the interpretation of the SFS.

R. We agree with the reviewer and the supplementary figure 4 actually shows the SFS distribution before the MAF filter, not only for the entire dataset but also for the subsets of the Mesoamerican and the Andean samples. We did not make a large analysis of the SFS because our previous publications (Chacon-Sanchez et al., 2017 and Garcia et al., 2021) included a complete analysis of the variability between and within the main Lima bean gene pools, which was based on GBS data for a larger number of samples, both total and per population. The SFS shown here serves mainly to confirm that the SNP dataset was sound. The MAF filter is only performed to eliminate for downstream analysis SNPs with alternative alleles present in less than 3 samples. This is a usual filtering technique to eliminate potential false SNPs resulting from remaining genotyping errors.

-Lines 186-188: I suggest applying another phylogenetic approach to construct the tree, such as the maximum-likelihood method used for the LTR transposons (FastTree).

R. We applied the FastTree method to construct a maximum likelihood tree from the data (see figure below). The resulting tree is consistent with the neighbor joining tree shown in Figure 2 and with the trees shown in Garcia et al., 2021. We wish to clarify though that we normally do not build phylogenetic trees from population genomics data within a species, because we believe that the distribution of variability within a species does not normally follow a tree structure. Hence, we refrain from interpreting the neighbor joining tree as a phylogenetic tree, and hence we do not use the word “phylogenetic” when we describe neighbor joining trees. The neighbor joining tree merely serves as a graphical display of the genetic distances that can be obtained from the genomic variation data. As a technical consequence of this argument, the FastTree software does not receive a matrix of genotyped SNPs, but a multiple sequence alignment of a region that is expected to have some level of conservation among samples (terminal inverted repeats in the case of LTRs). This is also the case for most of the well known

packages to build phylogenetic trees. To produce the tree included in this answer, we had to select a random subset of 20,000 SNPs and then build a mock multiple sequence alignment from these SNPs. Since we do not actually consider this as a recommended procedure, we decided not to include this result in the manuscript.

-Fig 2C-D): What does the salmon colour indicate? Are the values in the salmon outliers?

R. Correct, we used the salmon color to show the region of significance for each statistic. We updated the legend of the figure to explain this.

-Line 228: low genetics diversity and high Fst?

R. Correct. We fixed the sentence

-Lines 350-352: Why a peak close to 0.5 is expected? Adding the expected SFS would allow comparison if there is a deviation between the observed and expected pattern of the SFS. There are neutral and non-neutral evolutionary processes that can be affecting the SFS.

R. In this case we expected a peak at 0.5 because we expected that the SFS distribution derived from variable TEs would be similar to that derived from SNPs. The main evolutionary force producing this peak is high population structure, as it has been reported in Chacon-Sanchez et al. 2017 and Garcia et al. 2021. We improved the text to clarify that the expectation in this case is based on the SFS derived from SNPs.

*-Lines 367-368: What percentage of the *P. lunatus* genes presented TEs in their flanking regions?*

R. We calculated and reported this percentage (38%).

-Lines 579-581: What do the numbers after "domesticated-" and "wild-" mean? This is not clear to me.

R. They indicate the number of sequenced samples per population. We improved the sentence.

REVIEWERS' COMMENTS:

Reviewer #1 (Remarks to the Author):

Overall, I believe the authors did an excellent job of answering all questions and concerns. Their manuscript is clear, cohesive and better-organized now. I have no further concerns or comments and believe this will be add new knowledge to an important group of plants.

Reviewer #2 (Remarks to the Author):

I'd like to acknowledge the author for considering the comments and integrating suggestions. The quality of the manuscript and the figures has significantly improved. There are still some minor comments I'd like to address:

Previous comment: Lines 182-184 and Supp Fig. 4

R. We agree with the reviewer and the supplementary figure 4 actually shows the SFS distribution before the MAF filter, not only for the entire dataset but also for the subsets of the Mesoamerican and the Andean samples. We did not make a large analysis of the SFS because our previous publications (Chacon-Sanchez et al., 2017 and Garcia et al., 2021) included a complete analysis of the variability between and within the main Lima bean gene pools, which was based on GBS data for a larger number of samples, both total and per population. The SFS shown here serves mainly to confirm that the SNP dataset was sound. The MAF filter is only performed to eliminate for downstream analysis SNPs with alternative alleles present in less than 3 samples. This is a usual filtering technique to eliminate potential false SNPs resulting from remaining genotyping errors.

Lines 188-190: Thank you for your clarification about the SFS. It seems to me that the information regarding the MAF distribution, which was constructed pre-filtering, is not indicated in the text. I'd suggest including this.

Previous comment: Lines 186-188

R. We applied the FastTree method to construct a maximum likelihood tree from the data (see figure below). The resulting tree is consistent with the neighbor joining tree shown in Figure 2 and with the trees shown in Garcia et al., 2021. We wish to clarify though that we normally do not build phylogenetic trees from population genomics data within a species, because we believe that the distribution of variability within a species does not normally follow a tree structure. Hence, we refrain from interpreting the neighbor joining tree as a phylogenetic tree, and hence we do not use the word "phylogenetic" when we describe neighbor joining trees. The neighbor joining tree merely serves as a graphical display of the genetic distances that can be obtained from the genomic variation data. As a technical consequence of this argument, the FastTree software does not receive a matrix of genotyped SNPs, but a multiple sequence alignment of a region that is expected to have some level of conservation among samples (terminal inverted repeats in the case of LTRs). This is also the case for most of the well known packages to build phylogenetic trees. To produce the tree included in this answer, we had to select a random subset of 20,000 SNPs and then build a mock multiple sequence alignment from these SNPs. Since we do not actually consider this as a recommended procedure, we decided not to include this result in the manuscript.

Thank you for performing a maximum likelihood approach. As you mentioned, it is consistent with the Neighbor-joining tree. I'd like to respectfully disagree with some of the affirmations regarding the use of SNPs for phylogenetics (phylogenomics). SNPs are widely used to construct phylogenetic trees since

they are homologous sites and non-polymorphic loci, which are non-informative, are excluded. Concatenating the polymorphic sites does not represent an impediment to using SNPs for phylogenetic analysis. Thousand or even millions of SNPs can be included for phylogenetic inferences, or selecting a subset according to the goals of the study (target genes, genic regions, intergenic regions, etc.). Finally, the population structure is usually consistent with the general structure of the phylogenetic inferences even though mixed ancestry samples can be noisy. Because the goal of the study is not closely related to phylogenetics and the tree shown in Fig. 2B is very similar to the one constructed with FastTree, I consider that replacing the figure is not necessary.

Lines 340-341: The number of PAVs or any type of variation is related to how closely the samples are to the reference. In this case, because the reference genome is a domesticated Mesoamerican accession it is expected that fewer PAVs are identified within this group and the larger number be present in the most distant group to the reference (Andean pool). I suggest including this in the discussion.

Previous comments Lines 350-352

R. In this case we expected a peak at 0.5 because we expected that the SFS distribution derived from variable TEs would be similar to that derived from SNPs. The main evolutionary force producing this peak is high population structure, as it has been reported in Chacon-Sanchez et al. 2017 and Garcia et al. 2021. We improved the text to clarify that the expectation in this case is based on the SFS derived from SNPs.

Thank you for addressing my question. I agree with you that the frequency distribution of the PAVs (and SVs in general) reflects, at least partially, the population structure and other neutral processes. Despite this, it has been suggested that this type of variation tends to present greater effects on the phenotypes compared to SNPs because they are longer. Therefore, negative selection might be acting on them and an excess of low-frequency variants is expected. SVs are a source of adaptive variation but they tend to be deleterious. In this scenario, the SFS of PAVs could be different from the one derived from the SNPs and a low-frequency peak would be expected in the former.

Dear reviewers

Many thanks for your evaluation of our manuscript 'Selection signatures and population dynamics of transposable elements in Lima bean'. We performed additional data analysis and made the changes in the manuscript needed to address each comment. Please find our answers below for each specific comment. To facilitate the revision process, we marked in red the changes performed from the previous version of the manuscript.

Reviewer #1 (Remarks to the Author):

Overall, I believe the authors did an excellent job of answering all questions and concerns. Their manuscript is clear, cohesive and better-organized now. I have no further concerns or comments and believe this will add new knowledge to an important group of plants.

R. We thank the reviewer for the evaluation of our manuscript. We are glad to hear that all previous comments were properly addressed.

Reviewer #2 (Remarks to the Author):

I'd like to acknowledge the author for considering the comments and integrating suggestions. The quality of the manuscript and the figures has significantly improved. There are still some minor comments I'd like to address:

R. We thank the reviewer for the evaluation of our manuscript. We are glad to hear that most previous comments were properly addressed. We followed the suggestions included in this review, performing appropriate changes in the results and discussion. Please see below our specific answer to each comment.

Previous comment: Lines 182-184 and Supp Fig. 4

R. We agree with the reviewer and the supplementary figure 4 actually shows the SFS distribution before the MAF filter, not only for the entire dataset but also for the subsets of the Mesoamerican and the Andean samples. We did not make a large analysis of the SFS because our previous publications (Chacon-Sanchez et al., 2017 and Garcia et al., 2021) included a complete analysis of the variability between and within the main Lima bean gene pools, which was based on GBS data for a larger number of samples, both total and per population. The SFS shown here serves mainly to confirm that the SNP dataset was sound. The MAF filter is only performed to eliminate for downstream analysis SNPs with alternative alleles present in less than 3 samples. This is a usual filtering technique to eliminate potential false SNPs resulting from remaining genotyping errors.

Lines 188-190: Thank you for your clarification about the SFS. It seems to me that the information regarding the MAF distribution, which was constructed pre-filtering, is not indicated in the text. I'd suggest including this.

R. We improved the text in the results to highlight that the SFS were performed before filtering. The new text (lines 172-175) reads as follows:

“The minor allele frequency (MAF) distribution of the overall population, derived from the raw genotype calls, shows an excess of SNPs with high frequency of the minor allele (Supplementary Figure 4). This can be explained by the population structure of the sequenced samples.”

Previous comment: Lines 186-188

R. We applied the FastTree method to construct a maximum likelihood tree from the data (see figure below). The resulting tree is consistent with the neighbor joining tree shown in Figure 2 and with the trees shown in Garcia et al., 2021. We wish to clarify though that we normally do not build phylogenetic trees from population genomics data within a species, because we believe that the distribution of variability within a species does not normally follow a tree structure. Hence, we refrain from interpreting the neighbor joining tree as a phylogenetic tree, and hence we do not use the word “phylogenetic” when we describe neighbor joining trees. The neighbor joining tree merely serves as a graphical display of the genetic distances that can be obtained from the genomic variation data. As a technical consequence of this argument, the FastTree software does not receive a matrix of genotyped SNPs, but a multiple sequence alignment of a region that is expected to have some level of conservation among samples (terminal inverted repeats in the case of LTRs). This is also the case for most of the well known packages to build phylogenetic trees. To produce the tree included in this answer, we had to select a random subset of 20,000 SNPs and then build a mock multiple sequence alignment from these SNPs. Since we do not actually consider this as a recommended procedure, we decided not to include this result in the manuscript.

Thank you for performing a maximum likelihood approach. As you mentioned, it is consistent with the Neighbor-joining tree. I'd like to respectfully disagree with some of the affirmations regarding the use of SNPs for phylogenetics (phylogenomics). SNPs are widely used to construct phylogenetic trees since they are homologous sites and non-polymorphic loci, which are non-informative, are excluded. Concatenating the polymorphic sites does not represent an impediment to using SNPs for phylogenetic analysis. Thousand or even millions of SNPs can be included for phylogenetics inferences, or selecting a subset according to the goals of the study (target genes, genic regions, intergenic regions, etc.). Finally, the population structure is usually consistent with the general structure of the phylogenetic inferences even though mixed ancestry samples can be noisy. Because the goal of the study is not closely related to phylogenetics and the tree shown in Fig. 2B is very similar to the one constructed with FasTree, I consider that replacing the figure is not necessary.

R. We appreciate the follow up on this discussion. We will investigate in more depth the use of genome-wide SNPs to perform phylogenetic inferences. Since we agree that the goal of the study is not related to phylogenomics, we did not make changes in response to this comment.

Lines 340-341: The number of PAVs or any type of variation is related to how closely the samples are to the reference. In this case, because the reference genome is a domesticated Mesoamerican accession it is expected that fewer PAVs are identified within this group and the larger number be present in the most distant group to the reference (Andean pool). I suggest including this in the discussion.

R. We included this reasoning in the discussion. The new text (lines 492-494) reads as follows:

“Because our analysis is guided by a Mesoamerican reference genome, we identified fewer PAVs within Mesoamerican accessions compared to more distant Andean accessions.”

Previous comments Lines 350-352

R. In this case we expected a peak at 0.5 because we expected that the SFS distribution derived from variable TEs would be similar to that derived from SNPs. The main evolutionary force producing this peak is high population structure, as it has been reported in Chacon-Sanchez et al. 2017 and Garcia et al. 2021. We improved the text to clarify that the expectation in this case is based on the SFS derived from SNPs.

Thank you for addressing my question. I agree with you that the frequency distribution of the PAVs (and SVs in general) reflects, at least partially, the population structure and other neutral processes. Despite this, it has been suggested that this type of variation tends to present greater effects on the phenotypes compared to SNPs because they are longer. Therefore, negative selection might be acting on them and an excess of low-frequency variants is expected. SVs are a source of adaptive variation but they tend to be deleterious. In this scenario, the SFS of PAVs could be different from the one derived from the SNPs and a low-frequency peak would be expected in the former.

R. Thanks for this possible explanation of the SFS distribution of PAVs. We included this reasoning in the following sentence in the discussion (lines 494-498):

“The MAF distribution of PAVs did not show the high frequency peak observed in the MAF distribution derived from SNPs. A possible explanation for this behavior is that structural variants tend to be deleterious and hence they might be subject to negative selection, which produces an excess of low frequency alleles.”